# Concerns related to returning home to a "difficult-to-return zone" after a long-term evacuation due to Fukushima Nuclear Power Plant Accident: A qualitative study

**Tomoo Hidaka**⬚*, **Hideaki Kasuga, Takeyasu Kakamu, Shota Endo**⬚, **Yusuke Masuishi**⬚, **Tetsuhito Fukushima**

Department of Hygiene and Preventive Medicine, School of Medicine, Fukushima Medical University, Fukushima, Fukushima, Japan

⬚ These authors contributed equally to this work.
* thidaka@fmu.ac.jp

**Data Availability Statement:** There are ethical restrictions on sharing the present study data

## Abstract

This study aimed to identify concerns related to returning to the Nagadoro district of Iitate Village, Fukushima Prefecture, Japan, in 2023 among its residents as it is designated as a difficult-to-return zone after the Fukushima Daiichi nuclear power plant accident. The following four concerns were extracted from the interviews and qualitative analysis: "Difficulties in restarting/continuing farming," which represent the difficulties in making a living from agriculture due to the absence of family members and neighbors, and the insufficient radiation decontamination; "Discriminatory treatment of products and residents from villagers," which suggests the presence of discriminations that residents of Nagadoro district are eccentrics and its agricultural products should not be treated together with those of other districts in the Village due to the recognition that the district is severely polluted by radiation; "Shift of the responsibility of returning home from the country to residents by scapegoating," which is characterized by the shift of responsibility from the government to the residents, including harsh social criticism of the residents of Nagadoro district for not returning even though the evacuation order has been lifted, when it should have been the government's responsibility to recover the district to a habitable state; "Loss of options for continued evacuation," which is the loss or weakening of the position of residents of the Nagadoro district who continue to evacuate, and of administrative compensation, resulting from the legal change that they are no longer "evacuees" after the evacuation order is lifted. The findings of this study will provide a foundation for the support of residents of the Nagadoro district after lifting the evacuation order scheduled for the spring of 2023. The findings may be transferable to the residents of other difficult-to-return zones expected to be lifted after the Nagadoro district and to also a radiation disaster-affected place in the future.

publicly because of privacy reasons. Data are available from the Ethics Committee of the Fukushima Medical University (contact at fmurec@fmu.ac.jp, or +8124-547-1825) for researchers who meet the criteria for access to confidential data.

**Funding:** TH received the grant from Japan Society for the Promotion of Science for this study (JSPS KAKENHI, grant number 16K17338). The JSPS had no role in study design, data collection and analysis, decision to publish, or preparation of the manuscript.

**Competing interests:** The authors have declared that no competing interests exist.

## Introduction

The Fukushima Daiichi nuclear power plant (FDNPP) accident in Japan in March 2011 contaminated large areas of the Fukushima Prefecture with radioactive materials [1, 2]. To initiate radioactive decontamination work, the Japanese government has designated three evacuation zones based on their air radiation doses: the "difficult-to-return zones" with 50 mSv/year or more; the "restricted residence zones" between 20 and 50 mSv/year; and the "evacuation order cancellation preparation zones." With the progress in decontamination work, land pollution has improved. Thus, the evacuation order was lifted in April 2017 for restricted residence zones and evacuation order cancellation preparation zones [3]. Radioactive decontamination work in the difficult-to-return zone is still ongoing as of March 2022 and has not yet been completed [3]. Residents who evacuated from the difficult-to-return zones in seven municipalities in Fukushima Prefecture could not return to their hometowns legally.

Although the Nagadoro district in Iitate Village, Fukushima Prefecture, is designated as a difficult-to-return zone, the evacuation order in this district will be lifted in the spring of 2023, allowing residents to return to their homes [4]. Nagadoro, located about 33 kilometers northwest of the FDNPP (Fig 1), was a small district with 72 families as of 2010 before the FDNPP accident [5]. Even though this population is much smaller than the total number of evacuees, which has reached a maximum of 165,000 [6], the return of people to the Nagadoro district has become the subject of great public interest. The Japanese government plans to lift evacuation orders in all difficult-to-return zones within the 2020s and allow residents to return to their homes [7]. The Japanese government has a grave responsibility to restore the difficult-to-return zones to habitable places and to ensure the return of residents to their homes; Japanese citizens are closely watching the future of this plan [8, 9]. Although Nagadoro is a small community, its successful recovery and the return of its residents are a symbol of Japan's efforts to recover from the Great East Japan Earthquake and the FDNPP accident in March 2011.

By the spring of 2023, the residents will decide whether to return to their homeland, the Nagadoro district. Regarding the return of FDNPP accident evacuees, past studies have revealed that lack of employment opportunities, housing problems, lack of options for school and industry, poor shopping accessibility, health concerns about radiation, and the effects of radiation on future generations are inhibiting factors [10–12]. In contrast, attachment to one's hometown and economic affordability are promoting factors [13, 14]. Moreover, mental health problems are prevalent among returnees of evacuation from the FDNPP accident in other districts/areas in Fukushima Prefecture due to radiation anxiety, family separation, and social isolation, even after returning [15–17]. These previous studies suggest that there are problems to be solved both before and after returning to their hometowns; however, these previous studies may not necessarily apply to the case of the Nagadoro district. In 2023, the evacuation order of the Nagadoro district will be lifted 12 years after the FDNPP accident, and the district will be the first case of the full-scale lifting of the evacuation order among difficult-to-return zones [3]. Individuals have been prohibited from entering Nagadoro district in law for such a long period. This long-term prohibition may have affected the state of restoration of industry, landscape, and infrastructure in Nagadoro. Its residents may have concerns about returning differently from those of other districts/areas' residents, as examined in previous studies. In this situation, where previous studies are not necessarily relied upon, it would be reasonable to use qualitative methods that inductively collect and analyze data instead of quantitative methods [18].

Importantly, there is a previous example outside Japan where residents have had to evacuate for a long time due to radiation disasters: the Chernobyl nuclear power plant accident in 1986. By 1990, more than 350,000 people had been removed and resettled from the most severely contaminated areas of Belarus, Russia, and Ukraine [19]. Whereas, several Chernobyl residents

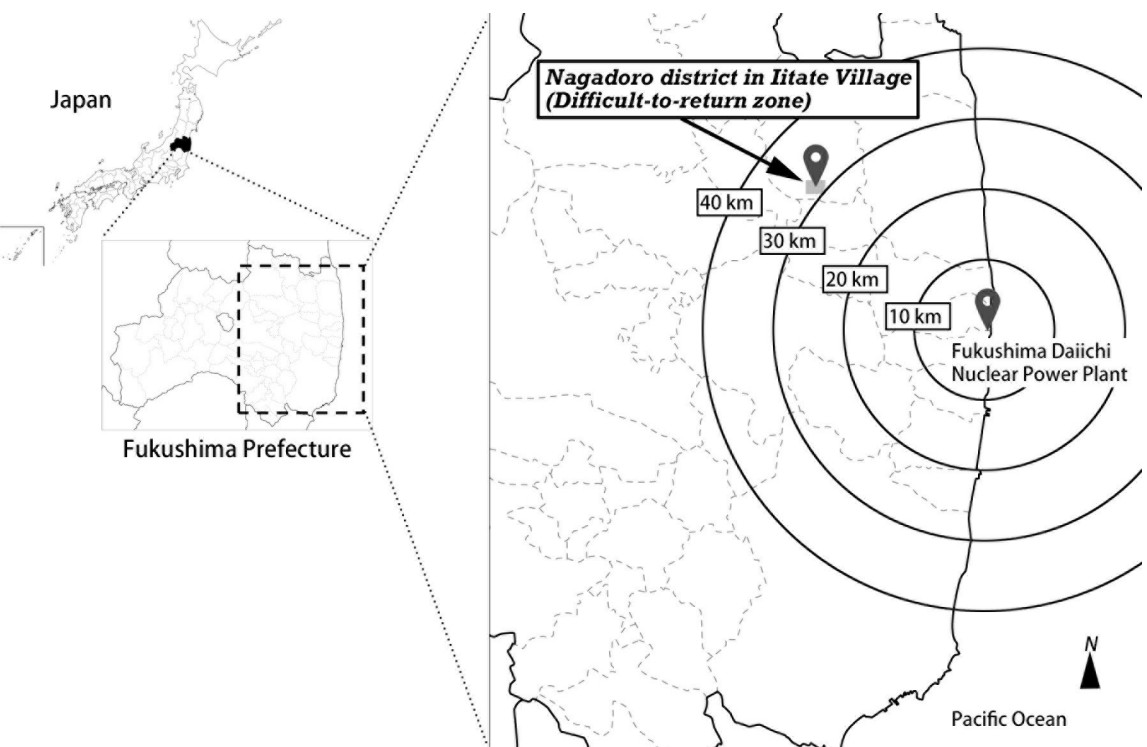

**Fig 1. Location of Nagadoro district in Iitate Village, Fukushima prefecture.** Concentric circles indicate the distance from the Fukushima Daiichi nuclear power plant.

have attempted to return to the Chernobyl area due to strong attachments to the place despite their fears and concerns on radiation [20]. This fact suggests that people forced to lose their homes due to the radiation disaster tend to have a strong desire to return; such a desire may be the same in both Chernobyl and Nagadoro district residents. We believe that identifying real concerns related to returning home of Nagadoro district residents contributes to determining the needed support and facilitating the recovery process in such a district, Chernobyl, and a place where a radiation disaster might happen in the future.

This study aims to identify the concerns related to residents returning to the Nagadoro district.

## Methods

### Research design and recruitment

Semi-structured interviews were conducted with one subject in this qualitative study. The interviewee, Mr. X, who was middle-aged and older male, was born and raised in the Naga-doro district of Iitate Village, Fukushima Prefecture, and worked in Iitate Village in his adult life. Although Mr. X evacuated to another location in Fukushima Prefecture than Iitate Village after the Great East Japan Earthquake, Mr. X continuously participated in community activities in the Nagadoro district. Mr. X obtained detailed information about the recovery status of the district and concerns about returning that the residents had. Because there were few potential interviewees due to the small population in the Nagadoro district, Mr. X was a valuable source of information on post and pre-return concerns held by the residents.

Regarding the sampling criterion, the subject(s) of qualitative research should be one or more individuals with rich experiences and information of interest to the researcher [21], and

a sample size of qualitative research is determined by the practical factors such as the depth and duration of the interview and what is feasible for an interviewer [22–24]. The Nagadoro district was originally a small community with only 72 households, and many of the residents had evacuated to other parts of Japan, leaving few potential interviewees. Under these conditions, where the possibility of recruiting interviewees was limited, it was practical to conduct multiple interviews with a tiny number of people, i.e., one person, to obtain thick data. This pragmatic choice of the sample, which emphasizes interview frequency over sample size, is often employed in qualitative research [25]. Mr. X appeared to meet all of these criteria of practicality and possession of rich information, and thus he was considered an appropriate interviewee.

For relationship establishment, the first and third authors had approached people from Iitate Village for a research project on the mental and physical health of FDNPP accident evacuees other than the present study, and thus both had known Mr. X since 2015. As that research project progressed, the authors were aware of Mr. X's background and knowledge regarding the Village, and Mr. X was familiar with the personalities, affiliations, and expertise of the first and third authors; it is reasonable to assume that rapport was formed. After receiving an explanation of the purpose of the present study, he readily agreed to be interviewed.

Note that detailed information about Mr. X, such as age and occupation, is confidential. The Nagadoro district was a community with a small population; thus, the interviewee can be easily identified based on such information. In addition, our analysis and results are meaningful regardless of such information. Therefore, we assumed that the confidentiality of such information was appropriate in terms of the anonymity of the individuals and methodological validity.

## Data collection

Three semi-structured face-to-face interviews were conducted on February 9, 2022, December 6, 2021, and November 11, 2021. Throughout these three interviews, the location was a conference room in Fukushima Prefecture.

Before these interviews, the media reported on a government policy that the evacuation order for the Nagadoro district would be lifted in the spring of 2023 [26, 27]. These reports were known to the residents of the Nagadoro district, including Mr. X. Thus, Mr. X was in a condition to easily represent concerns related to his return to the district. Therefore, the interviews followed a reasonable schedule.

The first author played the role of interviewer and asked the interviewee question, "After the evacuation order was lifted for the Nagadoro district, do you intend to return? What are the reasons for this?" This is the nature of the semi-structured interview that extracts a variety of topics, Mr. X mentioned concerns regarding both the cases: when returning and when not, in the form of a rich narrative. Repeating this question three times helped obtain detailed data on the conditions and background of residents' concerns, such as history and human relationships within the village and district.

Although the duration of the interviews varied according to Mr. X's schedule and was therefore inconsistent, the average duration was 67 minutes throughout the three interviews. The interviews were recorded using the IC recorder with the interviewees' permission, and a verbatim transcript was prepared for further analysis.

## Analytic procedure

The verbatim transcripts of the three interviews were combined and cross-sectionally analyzed using Steps for Coding and Theorization (SCAT), a coding method to summarize the

narratives inductively [28–30]. According to previous research, SCAT's analysis procedure consists of the following two steps: decontextualization, which generates themes from the text, and theorization, which summarizes the collected information to construct a theory [31]. Decontextualization consists of the procedures to segment and code sentences, breaking them down into smaller units such as noun phrases, and then labeling them using technical terms and academic concepts; the themes are generated through these abstraction procedures. Theorizing, a process of recontextualization, is a step, in which themes are integrated to organize the interviewees' perceptions and then summarize the findings of the analysis. In the theorizing process, an orderly storyline is reconstructed based on the extracted themes, and the storyline is then employed to describe suggestions that can be reasonably obtained from the data. Because of these careful procedures of gradually abstracting terms and elements from the text while maintaining correspondence with the original text explicitly and systematically, the SCAT has the following analytical advantages: high falsifiability [30] and the extraction of valuable findings from a small sample [32]. For a previous study using SCAT, anxiety over radiation among evacuees from an FDNPP accident was explored [33]. SCAT was expected to be an appropriate method for these methodological advantages and precedents in our study.

The first author conducted the analysis, and the second author reviewed and confirmed the results. These two authors were experienced researchers of qualitative methods. The analyses were performed using the SCAT coding form distributed by the developer of the SCAT [34] on Excel 2019 (Microsoft Corporation, Redmond, Washington, USA), and we exemplified the coding results to ensure transparency and trustworthiness of the analysis (S1 Table). The SCAT form consists of the following five coding steps for a text: 1. extracting noteworthy words or phrases from the text; 2. paraphrasing of step 1; 3. applying concepts from out of the text that account for step 2; 4. summarizing step 3 into themes and/or constructs in consideration of context; and 5. finding questions and tasks. As shown in this form, the SCAT employs a step-by-step abstraction procedure for coding. Finally, the creation of a storyline using the terms/phrases contained in step 4 and theoretical writing as an interpretation and discussion of the storyline is carried out.

Note that, due to the wide variety of the interviewee's narratives, we have created four storylines, named "content" one to four, inductively defined the expected cases of returning, and made discussions. The components of storylines are shown in the S2 Table.

### Ethics

The study protocol was approved by the Ethics Committee of the Fukushima Medical University, Fukushima, Japan (Application No. 2530). Written informed consent was obtained from the participant.

## Results and discussion

The interviewee was undecided about whether or not to return home, expressed possible life concerns on the cases of returning and not returning, and complained about the situation in which evacuees were forced to determine whether or not they returned. These results are summarized and described in five storylines, and their explanations and discussions are provided (Table 1).

### Content 1. Difficulties in restarting/continuing farming (returned case)

**The storyline.** *Nagadoro district is an agriculture-oriented area. Farming in the district is not solo work; which means, group farming by families and other groups takes place. However, given the original age structure of the community, which has many older adults and few young*

**Table 1. Concerns related to the returning Nagadoro district.**

| Expected cases | Content | Explanation |
|---|---|---|
| Returned | Difficulties in restarting/continuing farming | Difficulties in making a living from agriculture due to the absence of family members and neighbors, and the insufficient radiation decontamination. |
| | Discriminatory treatment of products and residents from villagers | Discriminations that residents of Nagadoro district are eccentrics and its agricultural products should not be treated together with those of other districts in the Village due to the recognition that the district is severely polluted by radiation. |
| Not returned | Shift of the responsibility of returning home from the country to residents by scapegoating | The shifting of responsibility from the government to the residents, including harsh social criticism of the residents of Nagadoro district for not returning even though the evacuation order has been lifted; the villainization of the residents who decided not to return, when it should have been the government's responsibility to recover the district to a habitable state. |
| Other (complaint on the situation) | Loss of options for continued evacuation | The loss or weakening of the position of residents of the Nagadoro district who continue to evacuate, and of administrative compensation, resulting from the legal change that they are no longer "evacuees" after the evacuation order is lifted. |

people, and the prospects that residents hardly return, a shortage of agricultural workers is inevitable. Even if the younger generation were to return, they are unskilled and inexperienced in agriculture due to failure or lack of successor training. Thus, the main population of farmers will be older adults. Although older adults may try to use information technology to improve their lack of workforce, they are not familiar with such technology. Even if the district was to adopt a cooperative farming system shared by several families, the lack of successful experience in such a system in Nagadoro might have a negative effect. Residents have tried to install such systems in the past. However, the attempts failed due to poor management and discord among residents; thus, residents may be uncomfortable with such a system. There is no way to resolve the difficulties in earning a livelihood.

The interviewee anticipated that if he and other residents of the Nagadoro district returned, they would not be able to farm, and would therefore have difficulty making a living. As a small mountain community, it is difficult to establish a prominent commercial presence in Nagadoro. Therefore, farming, the community's traditional occupation, would be their first choice of work upon their return. However, farming conditions were not met. Residents do not return without a job, and a job cannot be served without a returned population. The residents of Nagadoro district may be amid such a double-bind dilemma. Importantly, this pessimistic outlook on the impossibility of farming includes the sadness of losing a beloved homeland inherited from the district's ancestors, which is associated with high psychological distress [35–37]. Policy support is needed to achieve both business development to create conditions for returning to the land and to improve psychological damage done by lost attachments.

## Content 2. Discriminatory treatment of products and residents from villagers (returned case)

**The storyline.**   It is a well-known fact that the Nagadoro district has long been a difficult-to-return zone and that the reason for the evacuation designation is the severity of the radioactive contamination. Although radiation decontamination has been completed to some extent in the district, some places still need to be cleaned. Because decontamination of the mountains and forests around the district has not been carried out, radioactive materials are likely to seep into the ground. Thus, wells and river water are at a high risk of radioactive contamination. This situation highlights the difficulty of securing water sources, which are more necessary than anything else for agriculture. Although there are suitable and safe places for agriculture in the district, the people of other districts in Iitate Village recognize Nagadoro as a dangerous place for food production. Even if farming is resumed in the district, there will likely be conflicts and divisions

*within the villagers; people of other districts in the village may refuse to accept rice or any other product from Nagadoro. Moreover, people in other districts may consider Nagadoro residents as eccentrics living in such a dangerous place. The future will end with unsustainable agriculture, livelihoods, and depression without a solution to this discriminatory treatment of products/ residents.*

The interviewee was concerned that human relational problems might occur in the future, even if Nagadoro residents restarted farming and produced crops. The human relational problem here is discriminatory treatment due to doubts about the safety of the crops raised by people in other districts of Iitate Village. Nagadoro district includes areas suitable for crop production in terms of radiation level; however, the district is treated as if everything within the district has the same condition. Everything produced in the Nagadoro district is deemed radioactive, contaminated, and thus dangerous, as the residents may be eccentrics. These discriminatory viewpoints are prejudices that do not reflect the facts. Previous studies have indicated that such prejudice against certain minority groups is a social determinant of increased mortality [38, 39], and elimination of prejudice is one of the critical factors for both returning home and improving health among residents. Another study indicated that setting common goals for groups that cannot be achieved without cooperation will eradicate prejudicial views [40]. Therefore, the village/municipal government's initiative to set community goals that will bring the entire village together to work toward recovery and return may be effective for enhancing collaboration between the residents of Nagadoro and other districts and consequently eradicate such prejudice.

## Content 3. Shift of the responsibility of returning home from the country to residents by scapegoating (non-returned case)

**The storyline.**   *Decontamination work is a government project funded by taxes from Japanese citizens. From the government's point of view, the Nagadoro district is a model case or touchstone for promoting return, and the results of the decontamination project in the district will probably be evaluated in terms of the number of people who return. The duration of decontamination is predetermined; once such a process is completed, no additional decontamination may occur, even if the site remains contaminated and is thus unfit to live in. Once the schedule is digested, the fact that "decontamination has been completed" is created; it becomes a fait accompli. If most Nagadoro residents do not return to their homes despite the completion of radiation decontamination, a debate will occur in society. Decontamination is deemed as a waste of taxpayers' money, which in turn may lead to doubt among people in and outside of Iitate Village toward Nagadoro residents: "The decontamination of the Nagadoro district was funded by taxpayers' money and should have already been completed, so why don't the residents go back home?" Such doubts in society may worsen feelings toward Nagadoro residents. Although the government is supposed to be responsible for the FDNPP accident, radioactive contamination, and recovery, it seems that the residents of Nagadoro are being treated as villains or scapegoats. If the Japanese government recognizes that radiation decontamination will not be effective in increasing the number of people returning to their homes, based on the results of Nagadoro, such decontamination work in other difficult-to-return zones may be considered futile. There is a concern that Nagadoro's case and the choice of Nagadoro residents not to return may become an obstacle to decontamination and reconstruction efforts in other evacuation zones of Fukushima Prefecture.*

In this storyline, the interviewee expressed that the Nagadoro residents felt as if they were being forced to return because the government's goal was to increase the number of returnees, even though the government's radiation decontamination work was insufficient. Ideally, the

government should have taken responsibility for carrying out radiation decontamination work and improving the environment in the Nagadoro area to achieve conditions that the residents could live there. However, the government's efforts have been far from satisfactory.

The choice to return is attributed solely to the responsibility of the residents. In particular, the choice to not return may be considered a selfish decision. As indicated in the storyline, the term "villain" or "scapegoat" seems to aptly describe this situation. Scapegoating is defined as the act of assigning undue blame or punishment to a target for the negative outcome that results from other causes, to maintain or restore the individual's perceived control over the external world [41]. In short, the essence of scapegoating is to control thoughts, emotions, and/or impulses that make an individual feel anxious by making someone else the villain.

Therefore, improvement of the image of Nagadoro residents after this expected scapegoating may be achieved by reducing the feeling of anxiety among the public. The interviewee suggested that the Japanese public other than the Nagadoro district must be anxious because they do not know how long the decontamination process will take and how much money it will cost. The government may communicate the reestablished schedule and goals to the public and the residents of Nagadoro to clarify the prospects related to radiation decontamination work and reconstruction, thereby reducing public anxiety.

## Content 4. Loss of options for continued evacuation (other case: The complaint on the situation)

**The storyline.** *Radiation decontamination in the Nagadoro district was carried out only in and around the specified reconstruction and revitalization base designated by the Japanese government. Therefore, radiation levels may remain high near homes, farmland, and other places where the returnees dwell and/or use. Even though radiation levels in a part of the district may remain high, radiation decontamination work is considered complete, and the evacuation order is legally lifted. After lifting the evacuation order, the Nagadoro residents lost their status as evacuees, and the government's assistance they received as this status of evacuees ended. It is time that residents decide whether to continue their current life outside of their hometown or return home. However, neither of those choices is desirable. The passage of time is 12 years as of 2023 since the FDNPP accident and the long-term prohibition of entry due to its designation as a "difficult-to-return zone.' That makes it difficult for residents to return to their homes. Because of the significant deterioration and collapse of their former dwellings in the district, residents will have to incur the cost of rebuilding new dwellings to live in after their return. However, most residents cannot afford it. Without returnees, it would be impossible to rebuild and maintain the community. Although the conditions for returning to their hometown are not met, it is difficult to continue evacuating. This situation, in which residents are practically forced to return home, could be described as the government's abandonment policy. As previously noted, if the residents do not return, they will likely be treated as villains by the public. Rather than rushing the residents to choose whether to return or not, it is necessary to support a rational decision that whichever they choose, they will be able to live a stable life.*

In this storyline, the interviewee expressed that the choice to return was a cause for concern. The specified reconstruction and revitalization base set up by the Japanese government is aimed at creating new communities in tandem with the progress of reconstruction work at the expense of the Japanese government [42]. However, the radiation decontamination work conducted within this project is only carried out to a limited extent, such as in the vicinity of the base. Therefore, Nagadoro residents faced a difficult choice: return to the Nagadoro district, where life is difficult due to insufficient radiation decontamination, or move earnestly and make a new life away from their homeplace without adequate administrative compensation.

Tsujiuchi [43] pointed out that the setting of evacuation and returning orders without reference to an appropriate radiation dose after the FDNPP accident resulted in evacuees being forced to return and in an ongoing social division. Tsujiuchi [43] also asserted that these problems are 'structural violence'[44], which is indirect, built into the structure, and shows up as unequal power and, consequently, unequal life chances. The residents of Nagadoro seem to be in the middle of these structural problems. To improve this structural problem, it may be effective to allow evacuees the right to reserve judgment on their return and arrange for ongoing administrative support to be provided, even after the evacuation order is lifted.

## General discussion

This study explored the concerns related to returning home, held by residents of Nagadoro district, where the evacuation order is scheduled to be lifted in the spring of 2023, through interviews with a district resident. The results showed concerns about returning or not returning, and those about the situation itself in which the residents would have to choose whether to return. The concern about this situation could be extracted because this study adopted an exploratory and inductive approach using a qualitative method. The fact that the interviewee's narrative had variations, including the feeling of partially welcoming the return and concerns about future livelihoods, supports that our three interviews and interpretations were appropriate for transparency and trustworthiness.

Overall, the concerns seem to be consistent: the core and key factor in the solution of such concerns may be interpersonal relationships. If Nagadoro residents return to their hometown, they may face discrimination from residents of other village districts, and if they do not return, they may be scapegoated by the public and positioned as villains. Given such concerns about expected human relationship problems, it is rational that our interviewee problematized the situation of choosing to return or not to return itself.

The emphasis on human/social relationships in concerns related to returning home is noteworthy. Previous studies have indicated the inhibiting factors for returning home among FDNPP accident evacuees, such as lack of employment, housing, poor school and industry options, shopping convenience, and the health effects of radiation [12–14]. These inhibiting factors highlighted the personal and private aspects of returning home. In contrast, our study's results suggest that human, social, and public aspects were determinants of returning home among Nagadoro residents. This difference may stem from the following two characteristics: first, the lift of the evacuation order for the Nagadoro district is the latest among other evacuation zones, as shown by the fact that 12 years will have passed since the FDNPP accident when the order is lifted in spring 2023; second, the district has been designated as a difficult-to-return zone, which is a severely radiation-contaminated area.

The storylines generated by these narratives support these interpretations. In the case of a return, farming is not feasible because of the difficulty of securing a workforce (Content 1), and even if farming were to resume, there is a risk of discrimination against products and residents of the Nagadoro district by villagers (Content 2). In the case of non-return, there is a concern of scapegoating by the public (Content 3), even though it is reasonable for Nagadoro residents to not return. In the case of concern about the choice to return or not to return itself (Content 4), as pointed out by Tsujiuchi [43], the notion of "the setting of evacuation and returning orders without reference to an appropriate radiation dose" seems to have caused prolonged evacuation, difficulties in making appropriate return decisions according to the status of radioactive contamination in the district, and eventually, social division.

Importantly, these concerns related to returning home among Nagadoro residents will not be resolved by the efforts of individual residents; such concerns involve social and public

aspects. Such concerns should be addressed at the policy and administrative levels. The government's revision of the timetable for reconstruction projects and decisions on re-extending compensation to evacuees are concrete measures that can be taken. Japanese public's interest in the Great East Japan Earthquake and the FDNPP accident has declined rapidly [45], realizing further ongoing support measures to help evacuees, including Nagadoro residents, lead fulfilling lives.

After the lifting of the evacuation order in the Nagadoro district in the spring of 2023, the difficult-to-return zone will remain in parts of six municipalities in the Fukushima Prefecture (Minamisoma City, Tomioka Town, Okuma Town, Futaba Town, Namie Town, and Katsurao Village) [3]. Evacuation orders for the difficult-to-return zones of these municipalities will probably be lifted in the future. The residents of these remaining difficult-to-return zones may have the same concerns as those of the residents of the Nagadoro district in the period before and after the lifting of the evacuation order, as shown in the present study. The problem of discrimination among residents shown in this study is not limited to the case of the Nagadoro district but could occur after any future radiation disaster that may occur in Japan and worldwide. Therefore, although our interviewee was a resident of Nagadoro, we believe that our findings are transferable to future returning home-related concerns that may arise in the six municipalities mentioned above and a radiation disaster-affected place in the future. In light of the concept of "transferability" that corresponds to external validity in quantitative research and supports the quality of qualitative research, it is reasonable to assume that the present study had a novelty based on rich information for answering the purpose of the study [46], despite being a single-subject research design.

## Limitation

The present study could not analyze the temporal transition of the concerns held by residents of the difficult-to-return zone. By conducting a cross-sectional analysis, the present study succeeded in comprehensively extracting the concerns held by the interviewee. However, such concerns may change over time. In other words, the concerns held by residents and the corresponding support needed may fluctuate; the results of the past study of FDNPP accident evacuees with longitudinal qualitative methods support this view [47]. Longitudinal design studies, including qualitative and epidemiological studies, will be called upon to understand the details of such a transformation of concerns.

## Conclusion

This study aimed to identify concerns related to returning to the district in 2023 among residents of the Nagadoro district of Iitate Village, Fukushima Prefecture which is designated as a difficult-to-return zone after the FDNPP accident. The following four concerns were extracted using a qualitative analysis method, SCAT: "Difficulties in restarting/continuing farming," "Discriminatory treatment on products and residents from villagers," "Shift of the responsibility of returning home from country to residents by scapegoating," and "Loss of options for continued evacuation." The interviewee mentioned the case of returning or not returning to the district and was also concerned about the situation itself in which the residents had to choose whether to return. The findings of this study will provide a foundation for the support of residents of the Nagadoro district after lifting the evacuation order scheduled for the spring of 2023. The findings may be transferable to the residents of other difficult-to-return zones expected to be lifted after the Nagadoro district and to a radiation disaster-affected area in the future.

## Supporting information

**S1 Table. The SCAT coding form.**
(PDF)

**S2 Table. The components of storylines.**
(PDF)

## Acknowledgments

We would like to express our sincere gratitude to the interviewee for his cooperation with this study.

## Author Contributions

**Conceptualization:** Tomoo Hidaka, Takeyasu Kakamu.

**Data curation:** Tomoo Hidaka.

**Formal analysis:** Tomoo Hidaka, Hideaki Kasuga.

**Funding acquisition:** Tomoo Hidaka.

**Investigation:** Tomoo Hidaka.

**Methodology:** Tomoo Hidaka.

**Project administration:** Tetsuhito Fukushima.

**Resources:** Tomoo Hidaka, Takeyasu Kakamu.

**Supervision:** Tetsuhito Fukushima.

**Validation:** Hideaki Kasuga, Takeyasu Kakamu, Tetsuhito Fukushima.

**Writing – original draft:** Tomoo Hidaka.

**Writing – review & editing:** Takeyasu Kakamu, Shota Endo, Yusuke Masuishi, Tetsuhito Fukushima.

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
