## [Decision Letter · Decision Letter 0]

11 May 2022

PONE-D-22-10093Concerns related to returning home to a "difficult-to-return zone" after a long-term evacuation due to Fukushima Nuclear Power Plant accident: a qualitative studyPLOS ONE

Dear Dr. Hidaka:

Thank you for submitting your manuscript to PLOS ONE. After careful consideration, we feel that it has merit but does not fully meet PLOS ONE’s publication criteria as it currently stands. Therefore, we invite you to submit a revised version of the manuscript that addresses the points raised during the review process.

Concerns were raised about the case study aspect of the work, although this is not really clear from the paper.  Please address these in a revision as well as other concerns raised by the reviewers. Please submit your revised manuscript by Jun 25 2022 11:59PM. If you will need more time than this to complete your revisions, please reply to this message or contact the journal office at plosone@plos.org. Please include the following items when submitting your revised manuscript:A rebuttal letter that responds to each point raised by the academic editor and reviewer(s). You should upload this letter as a separate file labeled 'Response to Reviewers'.A marked-up copy of your manuscript that highlights changes made to the original version. You should upload this as a separate file labeled 'Revised Manuscript with Track Changes'.An unmarked version of your revised paper without tracked changes. You should upload this as a separate file labeled 'Manuscript'.

We look forward to receiving your revised manuscript.

Kind regards,

Gayle E. Woloschak, PhD

Section Editor

PLOS ONE

Journal Requirements:

Additional Editor Comments:

One reviewer had concerns about the work and noted that it does not fit PLoS objectives. Please attempt to address the concerns raised in the response to the reviews. Please address concerns raised by the other reviewer as well.

Reviewers' comments:

Reviewer's Responses to Questions

**Comments to the Author**

1. Is the manuscript technically sound, and do the data support the conclusions?

Reviewer #1: Partly

Reviewer #2: No

2. Has the statistical analysis been performed appropriately and rigorously? 

Reviewer #1: N/A

Reviewer #2: No

3. Have the authors made all data underlying the findings in their manuscript fully available?

Reviewer #1: No

Reviewer #2: No

4. Is the manuscript presented in an intelligible fashion and written in standard English?

Reviewer #1: Yes

Reviewer #2: No

5. Review Comments to the Author

Reviewer #1: This paper is about qualitative research on an evacuee’s concerns related to returning home after the Fukushima nuclear power plant accident. The authors interviewed one subject three times. In addition, the three verbatim transcripts were qualitatively analyzed using the Steps for Coding And Theorization (SCAT) method. It would be of interest to the readers working in the fields of qualitative researches and regulatory authority. Nevertheless, there are some comments I would like you to take into account:

The authors extracted four concerns from the three interviews and SCAT method. All the concerns seem to be feelings of anxiety. Are there no feeling of serenity due to returning home? The authors should add transparency and trustworthiness to the findings and interpretations of the three interviews.

Story-line in SCAT is defined “something which is written using all the codes in decontextualization in order to express meanings and significances which lie in data. Hence descriptions of coding and memoing should be given.

According to the consolidated criteria for reporting qualitative research (COREQ) checklist, the authors should introduce SCAT matrix form.

Page 6 line 2, Reference 21 is appropriate? Please confirm that.

Reviewer #2: This paper deals with the concerns of one disaster victim about returning to a municipality in the evacuation zone of the Fukushima Daiichi Nuclear Power Plant accident. This paper is on a single subject, and it is appropriate to consider it a case study. Plos one will not consder case reports. Plos one considers publishing qualitative research only if it adheres to appropriate study design and reporting guidelines. This study has not adequately considered sample size in qualitative research. In a single case, it may be possible to generate a hypothesis, but it is impossible to find a conclusion that is free from individuality.

6. PLOS authors have the option to publish the peer review history of their article (what does this mean?). If published, this will include your full peer review and any attached files.

Reviewer #1: No

Reviewer #2: No

---

## [Author Response · Author response to Decision Letter 0]

14 Jun 2022

Dear Reviewers and Editor,

We wish to express our appreciation to you for your insightful comments, which we believe have helped us to improve our manuscript. 

First, according to the editor’s comments, we have revised the style of manuscript and added the information regarding data disclosure to cover letter. The typos and grammatical errors were also corrected.

The revised text was highlighted in yellow for reviewers.

[Responses for Reviewer #1]

Q1. The authors extracted four concerns from the three interviews and SCAT method. All the concerns seem to be feelings of anxiety. Are there no feeling of serenity due to returning home? The authors should add transparency and trustworthiness to the findings and interpretations of the three interviews.

A1: Thank you for your suggestion. Some narratives partially welcomed the progress of reconstruction. For example, the interviewee expressed a somewhat positive feeling about the progress made in radiation decontamination work, even if some shortcomings remain, in the following sentence of the storyline of content 2: “Although radiation decontamination has been completed to some extent in the district, some places still need to be cleaned.” However, in our data, there was no all-encompassing affirmation or feeling of serenity in returning to the hometown. 

In light of the sentence “it is necessary to support a rational decision that whichever they choose, they will be able to live a stable life” in the storyline of content 4, the interviewee may have placed value on a sustainable life after the return rather than the return itself. Thus, concerns related to future life may have been expressed, whereas the feeling of serenity by returning home was not necessarily emphasized. Such an interpretation is rational in light of the current and anticipated future reconstruction status of the Nagadoro district, in which many residents will not return and the community will not work in future, as shown in the manuscript. 

As the reviewer noted, the descriptions mentioned above are essential in clarifying the transparency and trustworthiness of our data and interpretation. We have added the following sentence to the end of frist paragraph at General discussion section (lines 370-372, p. 16-17): “The fact that the interviewee's narrative had variations, including the feeling of partially welcoming the return and concerns about future livelihoods, supports that our three interviews and interpretations were appropriate for transparency and trustworthiness.”

Q2. Story-line in SCAT is defined “something which is written using all the codes in decontextualization in order to express meanings and significances which lie in data. Hence descriptions of coding and memoing should be given.

A2: The reviewer’s comment is correct. We present the SCAT coding form used in our analysis to the reader as a Supplement and attach it to the reviewer. Please find the attachment named “Supplement S1 Table. The SCAT coding form.” This form is provided free of charge online by Otani [A], the developer of SCAT, and allows for qualitative analysis using standardized procedures on Microsoft Excel. We have added the following underlined texts to the Analytic procedure subsection of Methods section to clarify the coding procedures in SCAT (lines 191-193, p.8): “The analyses were performed using SCAT coding form distributed by the developer of the SCAT [34] on Excel 2019 (Microsoft Corporation, Redmond, Washington, USA), and we exemplified the coding results to ensure transparency and trustworthiness of the analysis (S1 Table).”

Also added that (lines 193-200, p.8): “The SCAT form consists of the following five coding steps for a text: 1. extracting noteworthy words or phrases from the text; 2. paraphrasing of step 1; 3. applying concepts from out of the text that account for step 2; 4. summarizing step 3 into themes and/or constructs in considerations of context; and 5. finding questions and tasks. As shown in this form, the SCAT employs a step-by-step abstraction procedure for coding. Finally, the creation of a storyline using the terms/phrases contained in the step 4 and theoretical writing as an interpretation and discussion of the storyline is carried out.”

Note that due to the wide variety of interviewee's narratives, we have created four storylines, named "content" one to four, inductively defined the expected cases of returning, and made discussions. For readability, we presented a supplement file for the reader and the reviewer; please find the attached file regarding the final components of the storylines (“Supplement S2 Table. The components of storylines”). We have added the following text to the Analytic procedure subsection of Methods section (lines 201-203, p.8): “Note that due to the wide variety of interviewee's narratives, we have created four storylines, named "content" one to four, inductively defined the expected cases of returning, and made discussions. The components of storylines were shown in S2 Table.”

Q3. According to the consolidated criteria for reporting qualitative research (COREQ) checklist, the authors should introduce SCAT matrix form.

A3: The reviewer's comment is correct. The SCAT coding form, its explanation, and related texts shown in response to comment #2 is the answer to this comment #3. 

<Reference for Reviewer #1>

[A] Otani T. English Excel Form for SCAT. http://www.educa.nagoya-u.ac.jp/~otani/scat/scatform-eng.xls

[Responses for Reviewer #2]

We thank the reviewer for the comments. Since the comments consist of important questions regarding the quality of a qualitative study, we will respond to each one in detail.

Q1. Plos one will not consider case reports. Plos one considers publishing qualitative research only if it adheres to appropriate study design and reporting guidelines. This study has not adequately considered sample size in qualitative research.

A1: Our study design was a cross-sectional study using a qualitative method of longitudinal three interviews. There are examples of using such a research design in previous studies related to health and disasters [A-C], and these studies have yielded valuable results as a qualitative study. Similarly, it is rational to assume the appropriateness of our study in terms of study design. 

For reporting guidelines, our study conforms to the Consolidated criteria for reporting qualitative research (COREQ) guideline [D]. The COREQ mentioned the sample size in qualitative research as “Researchers should report the sample size of their study to enable readers to assess the diversity of perspectives included.” (p.356) However, this statement is not a criterion for determining adequacy based solely on sample size; it must be discussed how valuable the perspectives obtained are as findings in qualitative research. In addition, there is no mention of a sample size of qualitative research in the PLOS ONE instructions [E].

Importantly, in PLOS ONE, the studies by Guichot-Muñoz et al.[F] and Helgegren et al.[G] have been published as original articles with the explicit statement that they employed a "single case study design." Although the theme of these studies was a regional/community issue and thus the term "single case" meant a spatial location rather than a person, the fact of these publications indicates that PLOS ONE can accept a study with a single case. The fact that there are studies [H-I] in which the subject was one person related to mental health and psychological conditions published in other journals as the original article, not a case report, may support the appropriateness of our study.

From these discussions, it is rational that our study is potentially acceptable in PLOS ONE in terms of study design and research guideline compliance. Note that the value of our study as a qualitative study, despite the sample size of one person, is explained in the following responses to the reviewer’s comments.

Q2: In a single case, it may be possible to generate a hypothesis, but it is impossible to find a conclusion that is free from individuality.

A2: In qualitative research, past studies stated that statistical representativeness is not required and that sample size is determined by other conditions, such as the depth and duration of the interview and what is feasible for an interviewer [J-L]. Therefore, single-case qualitative studies can also be admitted as the original study [M]. The following three perspectives help discuss the appropriateness of a subject in a single-case qualitative study: the presence of in-depth information for answering the research question [N], transferability of the findings [O], and the frequency of interviews to a subject instead of the number of subjects [P].

For the in-depth information, we could collect critical information by conducting interviews with a subject familiar with the village's situation and history. As indicated in our manuscript, the interviewees had rich information about intentions of returning among a wide range of age groups in the village, the village's past human resource development, human relations among villagers, and the reconstruction policies. This richness of information contributed to the clarification of our study's objective/research question: What are the concerns among the Nagadoro district inhabitants about returning home? Thus, it is assumed that our interviewee was adequate as a sample.

Transferability is a concept used to assess whether the findings of a qualitative study can be generalized to other situations and people and is equivalent to "external validity" in quantitative research [O]. Whether the findings from single-sample qualitative study yield benefits that overcome individuality should be argued through discussion in terms of transferability; in other words, adequacy in sampling for qualitative research is not judged by the number of subjects per se [Q]. As we indicated in our manuscript, in addition to shedding light on the problems faced by the Nagadoro residents, our results may be applied to radiation hazards and their evacuees in other communities and regions that may arise in the future, as well as their recovery from such hazards. Thus, we believed that our findings had a transferability.

Regarding the emphasis on the frequency of interviews, a study that conducted a systematic review of sample sizes in interview studies is instructive [P]. In this paper, the frequency of interviews is employed as the "sample size" for examination instead of the number of people. This idea is that variation in topics is ensured by multiple interviews. In other words, the data can be considered of higher quality when one interviewee mentions a variety of content than when several interviewees say the same thing. This idea seems reasonable for the sampling strategy of our study, because the Nagadoro district was originally a small community with only 72 households, and many of the residents had evacuated to other parts of Japan, leaving few potential interviewees. Under these conditions, where the possibility of recruiting interviewees was limited, it was practical to conduct multiple interviews with a tiny number of people, i.e., one person, to obtain thick data. In fact, as indicated in the manuscript, we were finally able to obtain a wide variety of narratives from Mr X regarding the concerns upon returning home. Note that such sample decisions for pragmatic reasons related to the difficulty of accessing a particular study population are often acceptable in qualitative research [P].

The above descriptions should be presented to the reader, because such descriptions provide the basic direction for sampling/sample size of our study. We have added/revised the following texts to Research design and recruitment subsection at Methods section (lines 124-134, p. 5-6): “Regarding the sampling criterion, the subject(s) of qualitative research should be individual(s) who have rich experiences and information of interest to the researcher[21], and a sample size of qualitative research is determined by the practical factors such as the depth and duration of the interview and what is feasible for an interviewer[22-24]. The Nagadoro district was originally a small community with only 72 households, and many of the residents had evacuated to other parts of Japan, leaving few potential interviewees. Under these conditions, where the possibility of recruiting interviewees was limited, it was practical to conduct multiple interviews with a tiny number of people, i.e., one person, to obtain thick data. This pragmatic choice of the sample, which emphasizes interview frequency over sample size, is often employed in qualitative research[25]. Mr. X appeared to meet all of these criteria of practicality and possession of rich information, and thus he was considered an appropriate interviewee.”

Also added the following texts to the last paragraph of General discussion section (lines 417-420, p. 18-19): “In light of the concept of “transferability” that corresponds to external validity in quantitative research and supports the quality of qualitative research, it is reasonable to assume that the present study had a novelty based on rich information for answering the purpose of the study [46], despite being a single-subject research design.”

Again, the quality of qualitative research should be evaluated by the quality of information collection, transferability, and frequency of interviews, rather than by sample size itself. The fact that the first author has previously published a paper using multiple interviews with a person [R] may technically support that we were able to elicit rich information from a single interviewee in this study. Consequently, we are sincerely convinced that our paper meets the requirements for possible publication in PLOS ONE both theoretically and practically, and that its findings have transferability to contribute to the support of the affected population of the FDNPP accident and similar disasters in future.

<Reference for Reviewer #2>

[A] Edwards M, Wood F, Davies M, Edwards A. The development of health literacy in patients with a long-term health condition: the health literacy pathway model. BMC Public Health. 2012;12:130. 

[B] Baxter K, Glendinning C. Making choices about support services: disabled adults' and older people's use of information. Health Soc Care Community. 2011;19(3):272-279.

[C] Santaella-Tenorio J, Bonilla-Escobar FJ, Nieto-Gil L, Fandiño-Losada A, Gutiérrez-Martínez MI, Bass J, Bolton P. Mental Health and Psychosocial Problems and Needs of Violence Survivors in the Colombian Pacific Coast: A Qualitative Study in Buenaventura and Quibdó. Prehosp Disaster Med. 2018;33(6):567-574.

[D] Tong A, Sainsbury P, Craig J. Consolidated criteria for reporting qualitative research (COREQ): a 32-item checklist for interviews and focus groups. Int J Qual Health Care. 2007;19(6):349-357.

[E] PLOSONE. Submission Guidelines. https://journals.plos.org/plosone/s/submission-guidelines

[F] Guichot-Muñoz E, Balbás-Ortega MJ, García-Jiménez E. Literacy, power, and affective (dis)encounter: An ethnographic study on a low-income community in Spain. PLoS One. 2021;16(6):e0252782.

[G] Helgegren I, Rauch S, Cossio C, Landaeta G, McConville J. Importance of triggers and veto-barriers for the implementation of sanitation in informal peri-urban settlements - The case of Cochabamba, Bolivia. PLoS One. 2018;13(4):e0193613. 

[H] Fleming V. Hysterectomy: a case study of one woman's experience. J Adv Nurs 2003;44(6):575–82.

[I] Gullestad FS, Wilberg T. Change in reflective functioning during psychotherapy--a single-case study. Psychother Res. 2011;21(1):97-111.

[J] Field PA, Morse JM. Nursing research: the application of qualitative approaches. London: Chapman and Hall, 1989.

[K] Mays N, Pope C. Rigour and qualitative research. BMJ. 1995; 311: 109–112.

[L] Britten N. Qualitative Research: Qualitative interviews in medical research. BMJ. 1995;311(6999):251-253.

[M] Boddy CR. Sample size for qualitative research. Qual. Mark. Res. 2016; 19: 426-432.

[N] Mason J. Qualitative researching (2nd ed.). London: Sage publications. 2002.

[O] Carpentar C., Suto M. Qualitative research for occupational and physical therapists: a practical guide. Oxford: Wiley-Blackwell. 2008.

[P] Vasileiou K, Barnett J, Thorpe S, Young T. Characterising and justifying sample size sufficiency in interview-based studies: systematic analysis of qualitative health research over a 15-year period. BMC Med Res Methodol. 2018;18(1):148. 

[Q] Sandelowski M. Sample size in qualitative research. Res Nurs Health. 1995;18(2):179–183.

[R] Hidaka T, Kasuga H, Kakamu T, Fukushima T. Discovery and Revitalization of “Feeling of Hometown” from a Disaster Site Inhabitant's Continuous Engagement in Reconstruction Work: Ethnographic Interviews with a Radiation Decontamination Worker Over 5 Years Following the Fukushima Nuclear Power Plant Accident1. Jpn Psychol Res. 2021;63:393-405.

---

## [Decision Letter · Decision Letter 1]

15 Aug 2022

Concerns related to returning home to a "difficult-to-return zone" after a long-term evacuation due to Fukushima Nuclear Power Plant accident: A qualitative study

PONE-D-22-10093R1

Dear Dr. Hidaka,

We’re pleased to inform you that your manuscript has been judged scientifically suitable for publication and will be formally accepted for publication once it meets all outstanding technical requirements.

Kind regards,

Gayle E. Woloschak, PhD

Section Editor

PLOS ONE

Additional Editor Comments (optional):

Thanks for addressing concerns raised by the reviewers.

Reviewers' comments:

Reviewer's Responses to Questions

**Comments to the Author**

1. If the authors have adequately addressed your comments raised in a previous round of review and you feel that this manuscript is now acceptable for publication, you may indicate that here to bypass the “Comments to the Author” section, enter your conflict of interest statement in the “Confidential to Editor” section, and submit your "Accept" recommendation.

Reviewer #1: All comments have been addressed

Reviewer #3: (No Response)

2. Is the manuscript technically sound, and do the data support the conclusions?

Reviewer #1: Yes

Reviewer #3: Yes

3. Has the statistical analysis been performed appropriately and rigorously? 

Reviewer #1: N/A

Reviewer #3: N/A

4. Have the authors made all data underlying the findings in their manuscript fully available?

Reviewer #1: Yes

Reviewer #3: Yes

5. Is the manuscript presented in an intelligible fashion and written in standard English?

Reviewer #1: Yes

Reviewer #3: Yes

6. Review Comments to the Author

Reviewer #1: I have much pleasure in recommending this paper for publication. The manuscript has been substantially with changes highlighted point by point according to reviewers' comments. The authors revised and improved the manuscript as suggested.

Reviewer #3: This paper provides very insightful and valuable findings about the returning decision of the people having been evacuated from the nuclear disaster in Fukushima. However, the adequacy of single-case qualitative research on this study should be the critical issue, as reviewer #2 indicated in the previous round of the review.

I found the findings and conclusion drawn here is consistent to my personal observation in the field. On the other hand, I believe the four storylines shown here is shared with many residents not only in Nagatoro but also many area. In this sense, this conclusion may be derived even from multiple-case qualitative case study across several region in Fukushima, and which will provide more persuasive results for the reader.

Although I believer the single-case interview is not necessarily the best way to prove the result in this paper, I admit the method itself is scientific enough and result and conclusion is meaningful.

7. PLOS authors have the option to publish the peer review history of their article (what does this mean?). If published, this will include your full peer review and any attached files.

Reviewer #1: No

Reviewer #3: **Yes: **Shingo Nagamatsu

---

## [Editor Report · Acceptance letter]

18 Aug 2022

PONE-D-22-10093R1 

Concerns related to returning home to a "difficult-to-return zone" after a long-term evacuation due to Fukushima Nuclear Power Plant accident: A qualitative study 

Dear Dr. Hidaka:

I'm pleased to inform you that your manuscript has been deemed suitable for publication in PLOS ONE. Congratulations! Your manuscript is now with our production department. 

Kind regards, 

on behalf of

Dr. Gayle E. Woloschak 

Section Editor

PLOS ONE